# New insights into intranuclear inclusions in thyroid carcinoma: Association with autophagy and with *BRAF^V600E* mutation

**Suzan Schwertheim**[1]*, **Sarah Theurer**[1], **Holger Jastrow**[2], **Thomas Herold**[1], **Saskia Ting**[1], **Daniela Westerwick**[1], **Stefanie Bertram**[1], **Christoph M. Schaefer**[1], **Julia Kälsch**[1,3], **Hideo A. Baba**[1]*, **Kurt W. Schmid**[1]

**1** Institute of Pathology, University Hospital of Essen, University of Duisburg-Essen, Essen, Germany, **2** Institute of Anatomy and Electron Microscopy Unit of Imaging Center Essen, University Hospital of Essen, University of Duisburg-Essen, Essen, Germany, **3** Department of Gastroenterology and Hepatology, University Hospital of Essen, University of Duisburg-Essen, Essen, Germany

* hideo.baba@uk-essen.de (HAB); Suzan.Schwertheim@uk-essen.de (SS)

## Abstract

### Background

Intranuclear inclusions (NI) in normal and neoplastic tissues have been known for years, representing one of the diagnostic criteria for papillary thyroid carcinoma (PTC). BRAF activation is involved among others in autophagy. NI in hepatocellular carcinoma contain autophagy-associated proteins. Our aim was to clarify if NI in thyroid carcinoma (TC) have a biological function.

### Methods

NI in 107 paraffin-embedded specimens of TC including all major subtypes were analyzed. We considered an inclusion as positive if it was delimited by a lamin AC (nuclear membrane marker) stained intact membrane and completely closed. Transmission electron microscopy (TEM), immunohistochemistry (IHC), immunofluorescence (IF) and 3D reconstruction were performed to investigate content and shape of NI; *BRAF^V600E* mutation was analyzed by next generation sequencing.

### Results

In 29% of the TCs at least one lamin AC positive intranuclear inclusion was detected; most frequently (76%) in PTCs. TEM analyses revealed degenerated organelles and heterolyso-somes within such NI; 3D reconstruction of IF stained nuclei confirmed complete closure by the nuclear membrane without any contact to the cytoplasm. NI were positively stained for the autophagy-associated proteins LC3B, ubiquitin, cathepsin D, p62/sequestosome1 and cathepsin B in 14–29% of the cases. Double-IF revealed co-localization of LC3B & ubiquitin, p62 & ubiquitin and LC3B & p62 in the same NI. *BRAF^V600E* mutation, exclusively detected in PTCs, was significantly associated with the number of NI/PTC ($p = 0.042$) and with immunoreactivity for autophagy-associated proteins in the NI ($p \leq 0.035$). BRAF-IHC revealed that

**Data Availability Statement:** All relevant data are within the manuscript and its Supporting Information files.

**Funding:** The authors received no specific funding for this work.

**Competing interests:** The authors have declared that no competing interests exist.

some of these BRAF-positive thyrocytes contained mutant BRAF in their NI co-localized with autophagy-associated proteins.

## Conclusions

NI are completely delimited by nuclear membrane in TC. The presence of autophagy-associated proteins within the NI together with degenerated organelles and lysosomal proteases suggests their involvement in autophagy and proteolysis. Whether and how BRAF[V600E] protein is degraded in NI needs further investigation.

## Introduction

The existence of intranuclear inclusions (NI) in many normal and neoplastic tissues has been known for a long time, in diabetic patients [1] in hepatocytes [2, 3] and particularly in thyroid carcinoma [4, 5], where its presence is one of the diagnostic criteria for papillary thyroid carcinoma [4, 5]. Ultrastructural studies of hepatocytes revealed that the inclusions contained cytoplasmic structures, often with degenerative changes. This supports the assumption that the NI are entirely separated from the cytoplasm [6]. Two morphologically different types of NI can be distinguished. First, inclusions which are due to the accumulation of virus particles or glycogen and which are not membrane-bounded, and second, inclusions that are bounded by a nuclear membrane [7, 8]. Meningiomas showed NI resembling autophagic vacuoles with lysosomal bodies suggesting an active macroautophagy process [9].

In thyroid carcinoma (TC), however, NI were seen as invaginations of the cytoplasm, i.e. bordered by the nuclear membrane with a considerable variety of size and shape even in the same tumor [10]. Also, occasionally "nuclear bubbles", fixation artifacts, can be detected in TC which are distinguishable from inclusions by the lack of a surrounding nuclear envelope [10, 11]. There are several ultrastructural studies of inclusions in thyroid cancer [5, 12–14]. Söderström et al observed electron-dense spherical bodies in NI; Carcangiu and Oyama et al revealed that these inclusions contained cell organelles and were surrounded by a nuclear envelope. Oyama et al documented that these inclusions contained enlarged RER, many Golgi vesicles, small vesicles (diameter of 300–500 nm) and fragments of mitochondria or crumpled membranes caused by increased protein synthesis and/or protein accumulation. Autophagosomes with such crumpled membranes and abundant heterolysosomes, indicating degradation inside NI, were also detected [5]; Söderström et al also observed small vesicles in the inclusions [14]. Kaneko et al revealed that NI and nuclear grooves both were formed by the nuclear membranes [13].

Although there are many observations regarding NI, there is no study that addresses the issue of whether these NI have a biological function. Therefore, we have recently investigated NI in hepatocellular carcinoma and have shown that they contain autophagy-associated proteins and correlate with prolonged survival [8]. These inclusions were located entirely within the nucleus and could be developed by occlusion of cytoplasmic invagination. Since NI play an important role in the diagnosis of thyroid carcinomas, we now examined the inclusions in thyroid carcinoma in respect to a possible function. In addition, it is also documented that strong immunopositivity for ß-catenin was detected within NI in 83% of PTCs [15]. Thus, we performed 3D reconstruction of the nuclei to investigate the development of the inclusions in thyroid carcinoma. In order to elucidate whether autophagy-associated proteins [16] are

detectable in the inclusions, we examined them immunohistochemically for the presence of p62/sequestosome1, ubiquitin, LC3B, cathepsin B and cathepsin D.

The detection of *BRAF* Mutation is an important diagnostic tool for papillary thyroid cancer [17]. BRAF activation is involved in many biological processes including cell proliferation and autophagy [18, 19]. The aims of the present study were to describe in detail NI in thyroid carcinoma, clarify their biological function and to investigate a possible connection to the occurrence of *BRAF^V600E* mutation.

## Materials and methods

### Patients

For this study, 107 tissue samples of thyroid carcinoma (71 female, 36 male; mean age 48 years, range 9–84 years), including all major subtypes, were obtained retrospectively from the archives of the Institute of Pathology, University Hospital of Essen, Germany. In all cases uniformly processed formalin-fixed and paraffin-embedded (FFPE) material was available. Slides were prepared and stained with HE (haematoxylin and eosin) according to institutional standards. The tumors were diagnosed according to the current WHO-criteria [20] and classified according to the TNM-system (8th edition). Table 1 provides details on patient and tumor characteristics.

Written informed consent was obtained from every patient. The study protocol conforms to the ethical guidelines of the 1975 Declaration of Helsinki and was approved by the Ethics Committee (Institutional Review Board) of the University Hospital Essen (reference number: 16-6917-BO).

### Tissue microarray construction

To study the expression of selected candidate proteins by immunohistochemistry we used tissue microarrays (TMAs). TMA construction was performed as previously described [8]. In the case of histologically obvious tumor heterogeneity, areas with lowest degree of differentiation

**Table 1. Patient's characteristics: Number of cases, mean age and sex of each group of patients.**

| Tumor type | Number | Mean age (range) | Male:female |
|---|---|---|---|
| ATC | n = 10 | 69 (55–84) | 4:6 |
| PDTC | n = 19 | 48 (13–80) | 7:12 |
| MTC | n = 10 | 48 (12–63) | 4:6 |
| FTC | n = 16 | 50 (9–76) | 6:10 |
| NIFTP | n = 9 | 48 (28–68) | 2:7 |
| PTC (n = 43) | | | |
| - Solid variant | n = 4 | 32 (19–58) | 0:4 |
| - Tall cell variant | n = 17 | 49 (10–77) | 5:12 |
| - Follicular variant | n = 9 | 42 (25–57) | 3:6 |
| - Conventional variant | n = 11 | 43 (18–75) | 5:6 |
| - Hobnail variant | n = 1 | 28 | 0:1 |
| - Columnar cell variant | n = 1 | 33 | 0:1 |
| Total | n = 107 | 48 (9–84) | 36:71 |

*Abbreviations*: ATC = undifferentiated (anaplastic) thyroid carcinoma; PDTC = poorly differentiated thyroid carcinoma; MTC = medullary thyroid carcinoma; FTC = follicular thyroid carcinoma; NIFTP = non-invasive follicular thyroid neoplasm with papillary-like nuclear features; PTC = papillary thyroid carcinoma; n = number.

were selected. Three 1-mm-thick tissue cores were taken from each thyroid carcinoma specimen. Each TMA contained three corresponding tumor-free thyroid tissue cores as controls and cores with myocardium tissue for TMA orientation. Additionally, HE-staining was performed from each TMA.

## Immunohistochemistry

1 to 2µm thick sections from FFPE tissue blocks were cut, dewaxed and pretreated. Immunohistochemical (IHC) stainings were performed with an automated staining device (Dako Autostainer, Dako, Glostrup, Denmark). IHC was carried out with antibodies against ubiquitin (#Z0458, Dako), p62 (#sc-28359, Santa Cruz, CA, USA), LC3B (#0231–100, Nano Tools, Hamburg, Germany), cathepsin B (#sc-6490-R, Santa Cruz), cathepsin D (#sc-6486, Santa Cruz), lamin AC (#2032, Cell Signaling, Danvers, MA, USA) and BRAF<sup>V600E</sup> (#ab228461, Abcam, Cambridge, UK). Detailed information on used antibodies and staining protocols are given in S1 Table. Negative controls were included in every run. For negative controls, slides were incubated with non-immune immunoglobulin instead of the primary antibody, carried out in the same concentration as the primary antibody. For positive controls, tumor cases that presented with a specific staining during antibody establishment were included in every subsequent run.

One investigator (DW) evaluated the HE stained slides and the immunohistochemical stains. NI were counted in all three tissue cores per case and total numbers were normalized to one square millimeter.

## Next generation sequencing (NGS)

FFPE tumor tissue of all cases was characterized by NGS with the Illumina MiSeq sequencer (Illumina, San Diego, CA, USA) following manufacturer's instructions. After deparaffinization of FFPE tissue, DNA was extracted using an automated system and kit (Maxwell/RSC DNA FFPE Kit, Promega, Fitchburg, WI, USA) according to the manufacturer's instructions. DNA concentrations were determined by Qubit® 2.0 Fluorometer dsDNA HS assay kit (LifeTechnologies, CA, USA). All samples showed a concentration between 0.5–70 ng/µl.

A total amount of 45 ng DNA was used to perform multiplex-PCR (four primer pools with 10 ng/primer pool+ 10% excess). Multiplex PCR and purification were performed with the GeneRead DNAseq Custom Panel V2, GeneRead DNAseq Panel PCR Kit V2 (Qiagen, USA) and Agencourt® AMPure® XP Beads (Beckmann, USA), followed by measurement of total DNA amount by Qubit® 2.0 Fluorometer dsDNA HS assay kit. The library preparation was performed with NEBNext Ultra DNA Library Prep Set for Illumina (New England Biolabs, MA, USA), according to the manufacturer's recommendations by using 24 different indices per run. The pooled library was sequenced on MiSeq (Illumina; $2 \times 150$ bases paired-end run) and analyzed by Biomedical Genomics Workbench (CLC Bio, Qiagen, USA).

For targeted sequencing a customized thyroid-panel was designed containing regions of interest. The thyroid-panel contained 23 genes of the Wnt pathway and hot-spot regions out of the *BRAF*, *KRAS* and *NRAS* genes. The analyzed genes and exons are listed in S2 Table. The regions were covered by a total of 1135 amplicons. In all runs an average coverage of approximately 2500x was obtained.

## Transmission electron microscopy (TEM)

Fresh tissue from thyroid biopsy of a representative thyroid carcinoma patient was fixed in 2% glutaraldehyde in 0.1 M cacodylate buffer (cb), pH 7.3, for 4 h at room temperature (RT), washed in cb, post-fixed with 1% osmium tetroxide in cb, dehydrated in a graded series of alcohol and embedded in epoxy resin. We stained semi-thin sections with basic fuchsin and

methylene blue in order to define blocks of adequate quality. Ultrathin sections from selected blocks were mounted on copper grids, double-stained with uranyl acetate (1%) and lead citrate (0.4%) and examined using a Zeiss TEM 902A (Zeiss, Oberkochen, Germany). For digital image acquisition, we used an attached Morada slow-scan-CCD camera and the ITEM 5.2 software (both Olympus Soft-imaging-Systems, Münster, Germany).

### 3D imaging of immunofluorescence-labeled isolated nuclei

In order to visualize the location of the inclusion within the nucleus, 3D imaging on isolated complete nuclei was performed. We studied the spatial position of the inclusion in relation to the nucleus by scanning the whole nucleus. For immunofluorescence tumor cell nuclei of two representative thyroid patients were isolated. Paraffin sections (60 μm thick) were placed in a 1.5 ml reaction tube and deparaffinized in xylene. After removing the supernatant, the pellet was rehydrated in 99%, 96% and 70% ethanol and supernatant was discarded. To remove residual ethanol, the pellet was washed in Target Retrievel Solution pH9 (#S2367, Dako), centrifuged and the supernatant was removed. The pellet was resuspended in 400 μl Target Retrievel Solution pH9 and disrupted by a homogenizer. Heat-induced epitope retrievel (HIER) was performed by heating the suspension at 98 ˚C in a water bath for 60 min. After that, the reaction tube with the suspension was cooled at RT for 30 min. The protocol describing the isolation of nuclei with subsequent immunostaining is available at: http://dx.doi.org/10.17504/protocols.io.78phrvn. The suspension was centrifuged, supernatant was discarded and the pellet was washed with washing buffer (#WL583C2500; DCS, Hamburg, Germany). After centrifugation, the supernatant was discarded and the pellet was equilibrated in a antibody diluent composite with Dako REAL Antibody Diluent (#S2022, Dako) containing additionally 2% BSA and 0.5% saponin for the permeabilization of the cells for 30 min at RT. Immunostaining of the cells was performed in the 1.5 ml reaction tube by incubation with the primary rabbit monoclonal antibody against lamin AC (#ab193904, Abcam) diluted 1:50 in the antibody diluent composite overnight at 4 ˚C. Cells were centrifuged, washed and incubated with the secondary antibody Alexa Fluor 488-conjugated donkey anti-rabbit IgG (#A21206, Thermo Fisher Scientific, Waltham, MA, USA) in a dilution of 1:100 in the antibody diluent composite for 60 min at RT (S3 Table). After nuclear staining with DAPI (Sigma-Aldrich, Steinheim, Germany), cells were mounted in anti-quenching medium (Vectashield; Vector Laboratories, Inc. Burlingame, CA, USA). Microscopy was carried out using a Leica TCS SP8 STED confocal microscope (Leica Microsystems, Illinois, USA) and images were analyzed with the Application Suite X software (Leica Microsystems). For 3D reconstruction based on sectional images we used the open source software Fiji (ImageJ; www.fiji.sc) and z-stacks were deconvolved with the Huygens Professional software (Scientific Volume Imaging; www.svi.nl/ContactSVI).

### Double-labeling immunofluorescence studies of tissue sections

**Analysis of autophagy-associated proteins by double-immunofluorescence.** We performed double-immunofluorescence staining for LC3B with ubiquitin, p62 with ubiquitin and LC3B with p62 to investigate a possible co-localization of them in the same inclusion. FFPE tissue sections were cut at 1 μm, dewaxed, rehydrated and pretreated with Target Retrieval Solution (Dako) at pH 9.0 for 20 min at 97 ˚C. We used the primary antibodies anti-LC3B (#3868, Cell Signaling), anti-p62 (#BWL PW9860, Enzo, Life Sciences, NY, USA), anti-p62 (#Sc-28359, Santa Cruz) and anti-ubiquitin (#NB300-130, Novus, Littleton, CO). The secondary antibodies used were Cy3-conjugated goat anti-rabbit IgG (H+L) (#111-166-045, Dianova, Hamburg, Germany) and Alexa Fluor 488-conjugated goat anti-mouse IgG1 (#A21121, Thermo Fisher Scientific). Double-labeling for LC3B with ubiquitin: Anti-LC3B antibody (1:20) was labeled

with Cy3 (1:100) and anti-ubiquitin antibody (1:1000) with Alexa Fluor 488 (1:100), each incubation was performed twice for 30 min at RT. Double-labeling for p62 with ubiquitin: Anti-p62 (Enzo) antibody (1:500) was labeled with Cy3 (1:800) and anti-ubiquitin antibody (1:1000) with Alexa Fluor 488 (1:100). Double-labeling for LC3B with p62: Anti-LC3B antibody was labeled with Cy3 and anti-p62 (Santa Cruz) antibody with Alexa Fluor 488 (for details, see S4 Table). DNA was stained with DAPI and images were analyzed as described above.

**Double-immunofluorescence labeling of *BRAF^{V600E}* and autophagy-associated proteins.** To analyze whether mutant BRAF is co-localized with autophagy-associated proteins anti-BRAF^{V600E} (#ab228461, Abcam) was double-labeled with either anti-LC3B (#3868, Cell Signaling) or anti-p62 (#BWL PW9860, Enzo) or with anti-ubiquitin (#Z0458, Dako). FFPE tissue sections were treated as described above. We performed fluorochrome labeling of BRAF^{V600E} with the VectaFluor™ Excel Amplified DyLight® 594 Anti-Mouse IgG Kit (#DK-2594, Vector Laboratories). The Donkey Anti Rabbit Alexa Fluor 488 (#A21206, Thermo Fisher Scientific) was used for fluorochrome labeling of p62, LC3B and ubiquitin.

Briefly, tissue sections were incubated 2x with anti-BRAF^{V600E} antibody (1:100) for 60 min and afterwards with DyLight594 (ready to use) for 30 min at RT. Double-labeling of BRAF^{V600E} with LC3B: sections were incubated with LC3B (1:100) and afterwards with Alexa Fluor 488 (1:100). Double-labeling of BRAF^{V600E} with p62: sections were incubated with p62 (1:250) and afterwards with Alexa Fluor 488 (1:100). Double-labeling of BRAF^{V600E} with ubiquitin: sections were incubated with ubiquitin (1:500) and afterwards with Alexa Fluor 488 (1:100); detailed information is provided in S5 Table. DAPI was used for DNA staining and images were analyzed as described above.

## Statistical analysis

Statistical analysis was performed using the Statistical Package for Social Sciences (SPSS 24.0, Chicago, IL, USA). Relationships between categorical parameters were investigated using the two-sided Fisher's exact test. Further, we used Mann–Whitney *U*-test to assess whether the *BRAF* mutational status correlates firstly with the number of NI and secondly with the number of inclusions positively stained for autophagy-associated proteins; $p \leq 0.05$ was defined as statistically significant.

## Results

### Characterization of the intranuclear inclusions (NI)

We considered an inclusion as positive if it was delimited by a lamin AC (nuclear membrane marker) stained intact membrane and completely closed. Study of three 1-mm-thick tissue cores of each patient in the TMA sections of the thyroid cohort revealed evaluable material for lamin AC staining in 99/107 cases. Briefly, 29% of the thyroid carcinoma cases harbored at least one lamin AC positive inclusion. The maximum number of NI per case was 114/mm$^2$ (Table 2). HE-staining (left image, Fig 1A) demonstrates that these NI have different shapes, numbers and sizes. Lamin AC IHC staining (right image, Fig 1A) reveals the complete closure of the inclusion by the nuclear membrane with no contact to the cytoplasm.

### Detection of *BRAF^{V600E}* mutation

Analysis of the thyroid cohort by NGS revealed that 14 of 107 cases (13%) harbored the *BRAF^{V600E}* mutation, whereby it was exclusively present in 14/43 (32%) PTCs. We identified the *BRAF* mutation mainly in the PTC tall cell variant with 8 of 14 (57%) and in the conventional PTC with 6 of 14 (42%), regarding the number of BRAF-mutated cases. In *NRAS* the

**Table 2. Number of NI and cases with *BRAF$^{V600E}$* in the thyroid carcinoma cohort.**

| Tumor type | Mean NI number / mm$^2$ per case (range) | Number of cases with NI* (%) | n valid cases | Number of cases with *BRAF$^{V600E}$* | n valid cases |
|---|---|---|---|---|---|
| ATC | 0.10 (0–1) | 1 (10.0%) | 10 | 0 | 10 |
| PDTC | 0.12 (0–1) | 2 (11.8%) | 17 | 0 | 19 |
| MTC | 0.30 (0–3) | 1 (10.0%) | 10 | 0 | 10 |
| FTC | 0.25 (0–3) | 2 (12.5%) | 16 | 0 | 16 |
| NIFTP | 0.11 (0–1) | 1 (11.1%) | 9 | 0 | 9 |
| PTC | | | | | |
| - Solid variant | 1.75 (0–6) | 2 (50.0%) | 4 | 0 | 4 |
| - Tall cell variant | 14.79 (0–53) | 13 (92.9%) | 14 | 8 | 17 |
| - Follicular variant | 0.88 (0–7) | 1 (12.5%) | 8 | 0 | 9 |
| - Conventional variant | 22.11 (0–114) | 4 (44.4%) | 9 | 6 | 11 |
| - Hobnail variant | 2 | 1 (100.0%) | 1 | 0 | 1 |
| - Columnar cell variant | 22 | 1 (100.0%) | 1 | 0 | 1 |
| Total | 4.60 (0–114) | 29 (29.3%) | 99 | 14 | 107 |

*Abbreviations*: NI = intranuclear inclusions; n = number.

* Number of valid cases with the presence of at least 1 intranuclear inclusion

Q61R mutation was found in 3/107 (2.8%) ATCs and 2/107 (1.9%) PDTCs. We detected with a maximum of 3/107 (2.8%) cases a rather low incidence of mutations in the genes of the Wnt pathway and we did not find any associations between these mutations and the occurrence of NI. More detailed information is provided in Table 2 and S6 Table. All mutations found in Cosmic with the prevalence of minimum 5% are listed; mutations listed additionally in the Clinvar database are marked.

### Increased occurrence of lamin AC stained NI in PTC

Lamin AC positive inclusions occur most frequently (22/29; 76%) in PTCs; in the PTC tall cell variant 13 of 14 cases have at least one membrane-bounded inclusion (Table 2). In contrast, we hardly detected any NI in ATC and PDTC. This is in line with the literature documenting that NI are mostly present in PTC [21, 22].

### Accumulation of autophagy associated proteins in the NI

Analysis of the NI by immunohistochemistry demonstrated positive immunoreactivity for the autophagy-associated proteins LC3B, ubiquitin, cathepsin D, p62 and cathepsin B in 14–29% of the cases (Fig 1B and Table 3) with dot-like LC3B staining in the inclusions (Fig 1B) i.e. endogenous LC3B indicating the induction of autophagy [23]. Data for immunostaining was available in 96–100 of 107 cases due to lack of suitable materials in the other cases. We detected in 29/99 (29.3%) of the cases at least one membrane-bounded intranuclear inclusion with positive p62 immunostaining. Further in 20/98 (20.4%) of the thyroid carcinomas at least one membrane-bounded inclusion with positive immunoreactivity for ubiquitin was shown (Fig 1B and Table 3); more details are listed in Table 3.

### Degenerative material accumulates in the NI with no connection to the cytoplasm

We performed TEM to study the NI in thyroid carcinomas in more detail. These inclusions are filled with degenerative materials such as remnants of mitochondria, endoplasmic

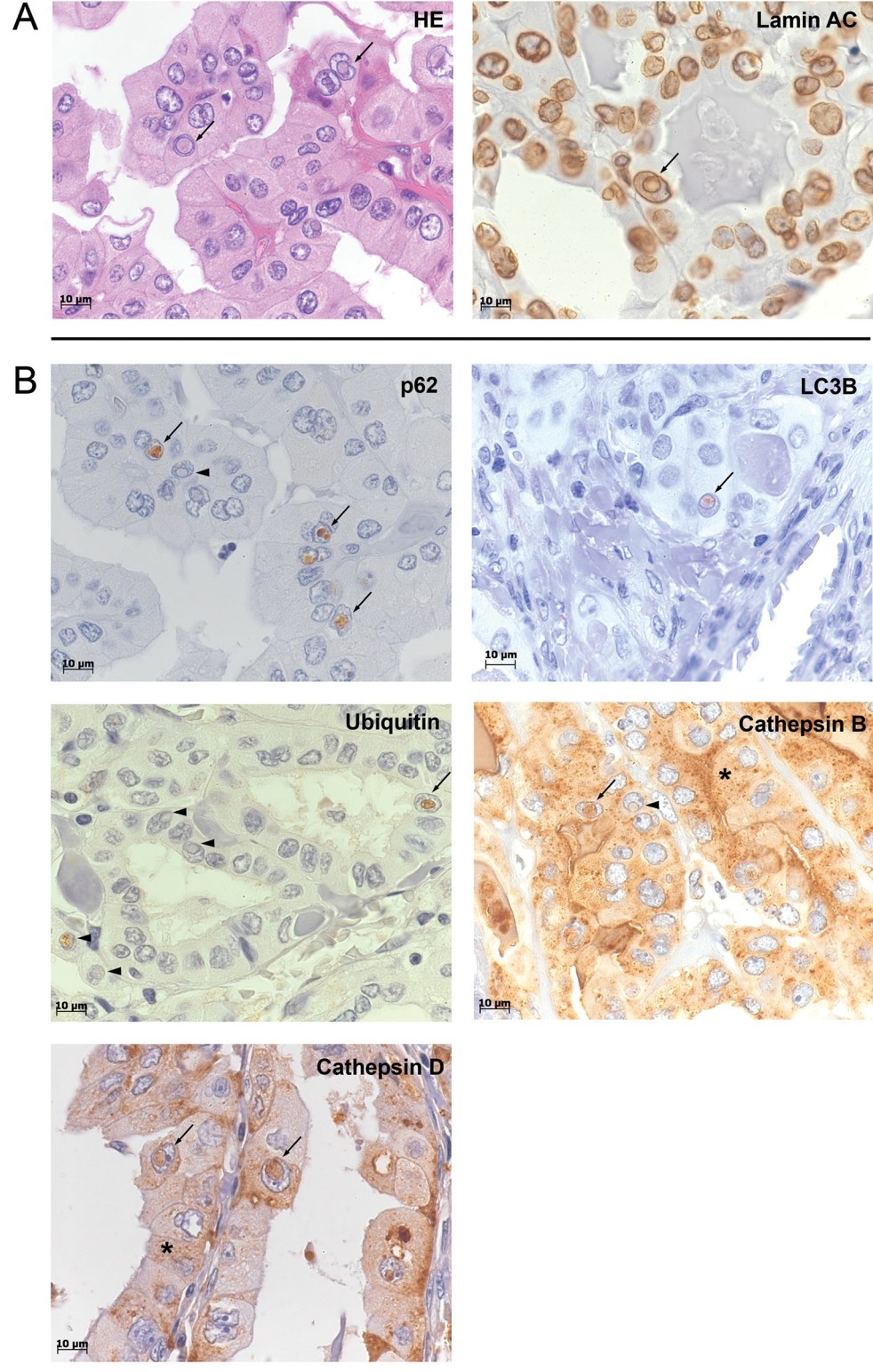

**Fig 1. Intranuclear inclusions in thyroid carcinoma (TC).** (A) Characterization of intranuclear inclusions (NI). Images show NI in representative cases of our TC cohort. HE-staining (left image) demonstrates that NI (arrows) are completely closed with no contact to the cytoplasm on this plane; positive lamin AC immunostaining (right image) of the limiting membrane of the inclusion (arrow) reveals its nuclear membrane origin. (B) Positive immunoreactivity for autophagy-associated proteins in NI. Immunohistochemistry (IHC) shows NI with varying degrees of

immunoreactivity for p62, LC3B, ubiquitin, cathepsin B and cathepsin D; partially different staining intensity in the inclusions for the same protein was detected. Image p62 demonstrates both NI with strong and partially dot-like immunoreactivity (arrows) and NI with almost no immunoreactivity (arrowhead). Also for cathepsin B and ubiquitin inclusions with different degrees of immunostaining were detected. Additionally, punctate distribution of LC3B in the inclusions (arrow) was observed. Cathepsin B- and cathepsin D-IHC also display cytoplasmic staining (asterix). Arrows: NI with strong immunoreactivity; arrowheads: NI with weaker or lacking immunoreactivity; asterix: cytoplasmic immunoreactivity. Original magnifications: 1,000 X.

reticulum, lamellar bodies and heterolysosomes (Fig 2Aii, 2Bi, 2C and 2Di). The absence of such degenerative changes in the cytoplasm indicates that these degradation processes take place completely delimited in the inclusions with no connection to the cytoplasm. Further, the content of the inclusions appears to be more condensed than cytoplasm (Fig 2A–2D). In fact even different stages of degradation of included material can be observed in electron microscopy, which resemble those of heterolysosomes when becoming residual bodies (Fig 2C). The inclusions are lined by the two membranes of the nuclear envelope with attached heterochromatin (h in Fig 2Ai, 2Aii and 2Di); along the inner membrane of the inclusion few ribosomes can be detected like the outer membrane of the nuclear envelope (Fig 2Aii). We observed nuclei with unusual invaginations (Fig 2A) and invaginations of surrounding cytoplasm into the nucleus (Fig 2D and 2Di). Thus, we suggest that some of these NI are formed by invaginations of the nuclear membrane.

## Inclusions are located completely in the nucleus and laminated by lamin AC

To investigate the shape of the NI and to clarify if they are merely nuclear membrane invaginations with persistent cytoplasmic contact or "real" inclusions, we performed 3D nuclear imaging with lamin AC immunofluorescence. 3D-reconstructions were performed for tumor cell nuclei of two thyroid patients. Two representative reconstructed NI are shown in Fig 3. Here we prove that there is no connection between the content of the NI and the cytoplasm in the studied section planes (Fig 3). Since the limiting membrane of the inclusions was lamin AC positive, it is most probable that it originates from the nuclear membrane. These results demonstrate that membrane-bounded NI are "real", exclusively intranuclear compartments completely separated from the cytoplasm (Fig 3).

## Correlation of ubiquitin with p62, LC3B, cathepsin B and cathepsin D

We used chi-square cross table analysis to study associations between p62, ubiquitin, LC3B, cathepsin B and cathepsin D immunostaining reactivity in the NI (Table 4). Samples were

**Table 3. Immunostaining of autophagy-related proteins in the intranuclear inclusions of the complete TC cohort.**

| Antibody | Thyroid carcinomas |
|---|---|
| | n/n valid cases (%) |
| Cathepsin D | 14 / 96 (14.6%) |
| LC3B | 20 / 100 (20.0%) |
| Ubiquitin | 20 / 98 (20.4%) |
| Cathepsin B | 20 / 97 (20.6%) |
| p62 | 29 / 99 (29.3%) |

The number of cases containing at least one intranuclear inclusion with positive immunoreactivity is shown;
n = number

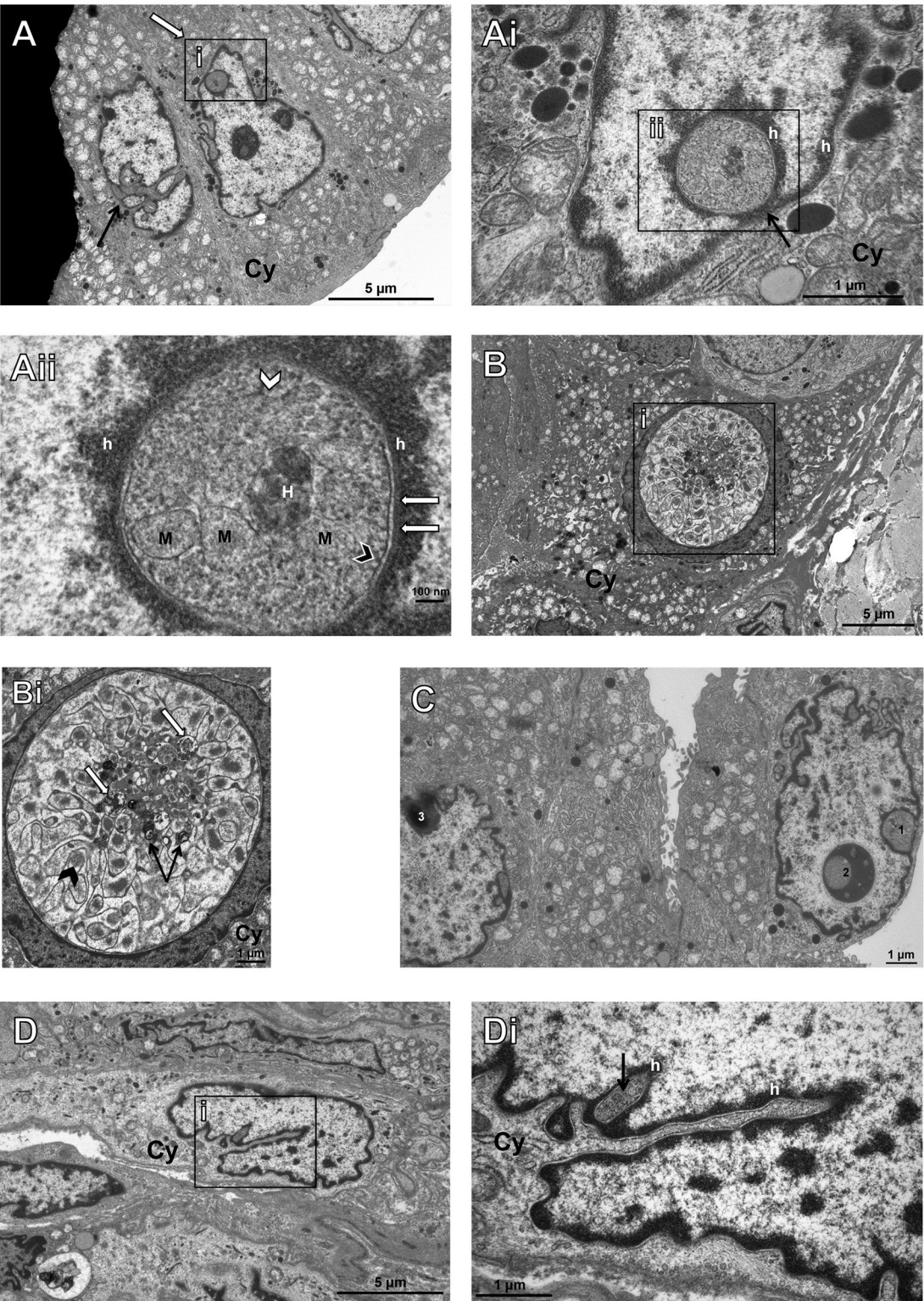

**Fig 2. Ultrastructural analysis of intranuclear inclusions in papillary thyroid carcinomas with transmissionelectron microscopy (TEM).** The image shows a nucleus with unusual invaginations (arrow) and another one with an additional intranuclear inclusion. (white arrow) (**A**). Increased magnification depicts the closure of the intranuclear inclusion with no contact to cytoplasm (Cy) (arrow) in the plane shown. This inclusion is lined by the two (inner and outer) nuclear membranes with attached membrane-associated heterochromatin (h) bordering a perinuclear space of regular width. There is no obvious

difference to the situation at the nuclear membrane, which surrounds the entire nucleus (**Ai**). At both locations few ribosomes (arrowhead) are attached to the outer nuclear membrane whereas a nuclear lamina (white arrows) is adjacent to the inner membrane. The detail shows a heterolysosome (H), degenerating mitochondria (M) with already disintegrated cristae and remnants of endoplasmic reticulum (white arrowhead) (**Aii**). Image **B** depicts a nucleus with an extraordinarily large inclusion, which is shown enlarged on the right (**Bi**). This inclusion contains an accumulation of heterolysosomes (white arrows), lamellar bodies (arrows) and punctual accumulations of fine granular moderate electron-dense material (arrowhead) in a fine matrix of low electron density rich in membranes (**Bi**). The next image (**C**) indicates degradation of material in NI. While the right intranuclear inclusion (1) shows a clear and well preserved bordering nuclear membrane, condensation digestion and accumulation of already degraded material can be seen in 2, which is also surrounded by a membrane. However, the latter entirely lacks attached heterochromatin and pores. Morphology of 3 resembles a residual body and in our eyes represents the final stage of degradation with extremely condensed material inside an inclusion, as if it could become ejected soon. Image **D** demonstrates a nucleus with very deep but thin invaginations of surrounding cytoplasm (**D**) and an inclusion containing condensed more electron-dense material (arrow). The inclusion is lined by the double membrane of the nuclear envelope (**Di**).

classified as positive (1) if at least one membrane-bounded inclusion showed positive immunoreactivity; cases lacking stained inclusions were classified as negative (0). Regarding the whole cohort, examination of 20 cases with positive ubiquitin immunostaining demonstrated that 19 of them also had NI with positive p62 immunostaining (p≤0.001) and 16 showed LC3B immunoreactivity (p≤0.001). We detected a significant relationship between ubiquitin and cathepsin B and cathepsin D immunoexpression in the NI (p≤0.001). Further, 18 of the 20 cases with LC3B- stained NI were also positive for p62 (p≤0.001) and 13 of the 14 cases

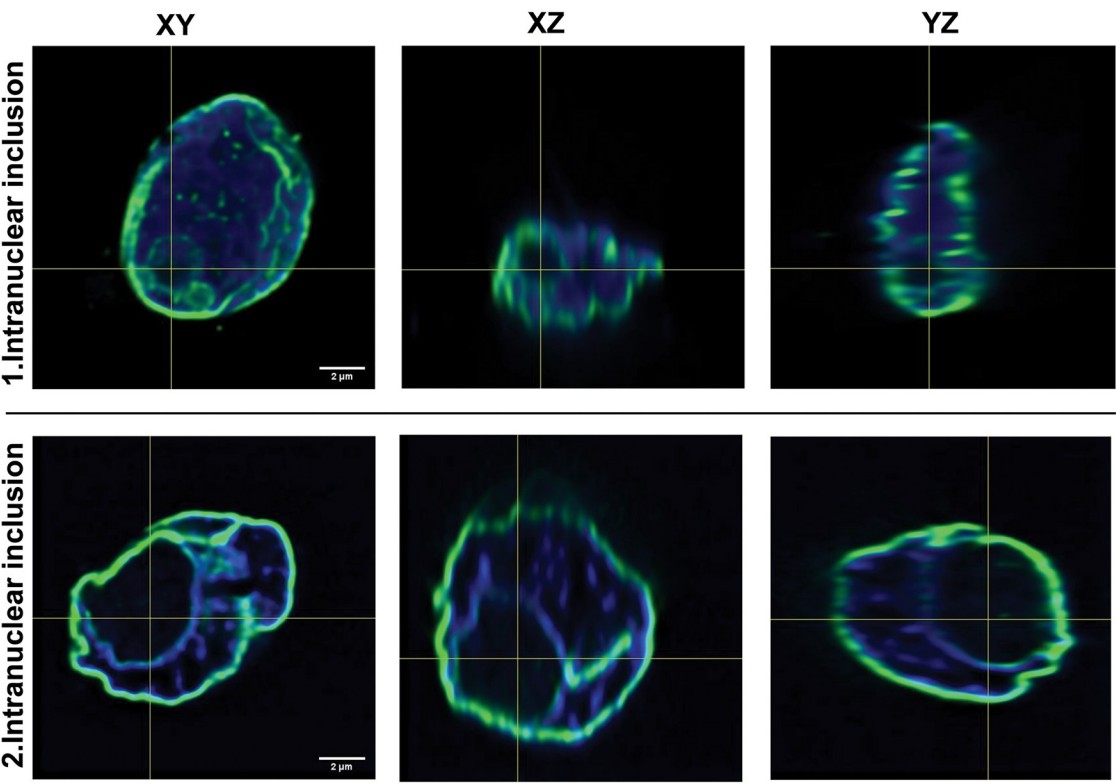

**Fig 3. 3D nuclear imaging with lamin AC immunofluorescence in X-, Y- and Z-axis.** For the first intranuclear inclusion 197 optical sections of 0.1μm were imaged and 137 of them were used for 3D reconstruction. The second intranuclear inclusion was reconstructed from 124 of 191 optical sections (0.1μm). Images reveal two representative tumor cell nuclei of a thyroid carcinoma patient. Both the nuclear membrane and the limiting membrane of the inclusion show lamin AC immunoreactivity (green staining), however that of the second inclusion is weaker. DNA staining was performed with DAPI (blue), which is lacking within the NI. The images demonstrate NI that are in all three planes completely closed and delimited by the nuclear membrane.

**Table 4. Associations between p62, ubiquitin, LC3B, cathepsin B and cathepsin D immunoreactivity in intranuclear inclusions.**

| Cross Tabs | | Whole TC cohort | | PTC cases | |
|---|---|---|---|---|---|
| Antibody | | n | P value | n | P value |
| Ubiquitin | p62 | 19 / 97 | < 0.001 | 17 / 37 | < 0.001 |
| | LC3B | 16 / 98 | < 0.001 | 15 / 37 | < 0.001 |
| | Cathepsin B | 16 / 97 | < 0.001 | 15 / 36 | < 0.001 |
| | Cathepsin D | 14 / 95 | < 0.001 | 14 / 36 | < 0.001 |
| p62 | LC3B | 18 / 99 | < 0.001 | 15 / 37 | 0.001 |
| Cathepsin B | Cathepsin D | 13 / 95 | < 0.001 | 13 / 36 | < 0.001 |

n = number of thyroid carcinomas with positive intranuclear inclusions/valid cases. P values were calculated using two-sided Fisher's exact test.

with positive cathepsin D immunostaining in the membrane-bounded NI revealed an additional immunoreactivity for cathepsin B ($p \leq 0.001$). Similar results for the PTC cases are detailed in Table 4.

### Autophagy-associated proteins are co-localized in the inclusions

To verify if these autophagy-associated proteins are located within the same inclusion, double-immunofluorescence labelled serial sections were analyzed by STED microscopy. Fig 4 shows three DAPI-stained nuclei with NI. Co-localization of ubiquitin/LC3B (Fig 4A), ubiquitin/p62 (Fig 4B) and LC3B/p62 (Fig 4C) is demonstrated in the same inclusion by the merged color yellow (Fig 4, arrows).

### Accumulation of mutant BRAF protein within the NI

We analyzed the FFPE thyroid carcinomas on mutant BRAF by IHC and confirmed our results of the NGS analysis (see above). Briefly, the same 14 of 107 cases, which were formerly proven to harbor the *BRAF^V600E* mutation, showed positive immunostaining for anti-B-Raf antibody, detecting exclusively BRAF^V600E. In 8 of 14 cases immunoreactivity for mutant BRAF was only seen in the cytoplasm but not in the inclusions. Intriguingly, the other 6 cases revealed, additional to cytoplasmic staining, positive BRAF^V600E staining within the NI showing an accumulation of BRAF^V600E protein in 0–28% of the inclusions. (Fig 5A).

### Co-localization of mutant BRAF with p62, LC3B and ubiquitin in the NI

To investigate if the accumulation of mutant BRAF protein is associated with autophagy and with proteolytic processes, we performed double-IF studies. STED microscopy images demonstrated three nuclei with accumulations of mutant BRAF within the inclusions (Fig 5B–5D). We detected co-localization of BRAF^V600E with LC3B (Fig 5B), with p62 (Fig 5C) and with ubiquitin (Fig 5D) respectively in the same inclusion. The formation of the merged color yellow proves the co-localization (Fig 5, arrows). Additionally we searched for a possible co-localization of mutant BRAF with autophagy-associated proteins in the cytoplasm. Double-IF studies showed few co-localizations of BRAF^V600E with ubiquitin and with LC3B but not with p62 after detailed z-stacks analysis; ubiquitin and LC3B immunostainings in the inclusions were much more intense than in the cytoplasm (S1 Fig).

### Increased occurrence of NI in PTC cases harboring *BRAF^V600E* mutation

As we detected *BRAF^V600E* mutation only in the PTC cases of our cohort, we asked whether *BRAF^V600E* mutation is somehow associated with NI. Mann–Whitney U-test

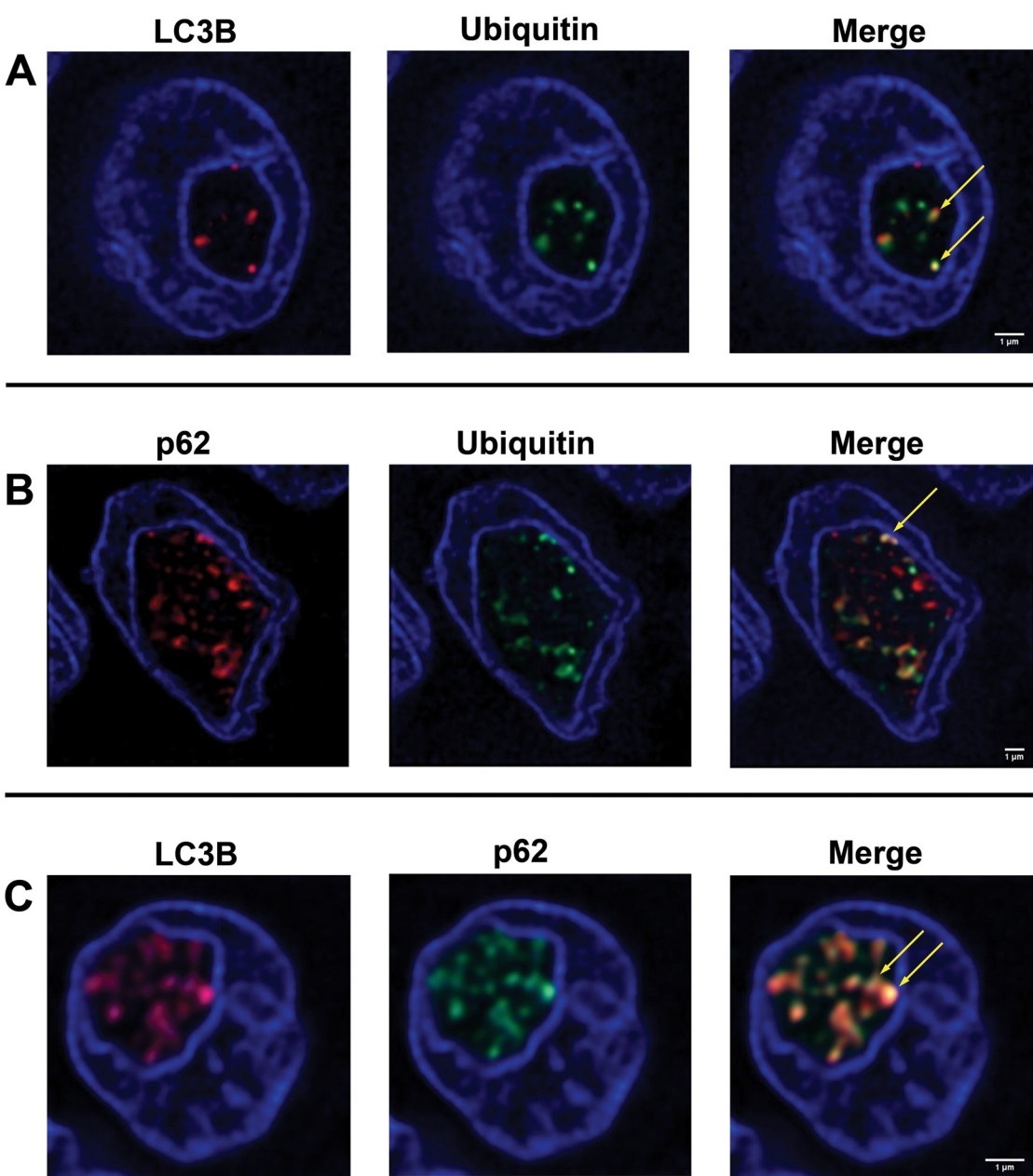

**Fig 4. Double-IF studies of autophagy-associated proteins in PTC sections. (A)** LC3B/ubiquitin double-IF labeling. DAPI staining (blue) reveals a nucleus with an intranuclear inclusion containing LC3B (red) and ubiquitin (green) immunostaining. The superimposition of both signals results in yellow color (arrows, merged image) proving a co-localization of the two proteins. **(B)** p62/ubiquitin double-IF labeling. An intranuclear inclusion with tense accumulation of p62 (red staining) and ubiquitin (green staining) is seen; yellow staining (arrow) in the merged image indicates p62/ubiquitin co-localization. **(C)** LC3B/p62 double-IF labeling. The images show strong LC3B (red staining) and p62 (green staining) immunoreactivity in the inclusion. There is strong LC3B/p62 co-localization displayed by the intense yellow color in the merged image (arrows).

showed a significant positive correlation between lamin AC stained NI and *BRAF* mutational status in PTC. Our results reveal that the number of NI is significantly higher in PTC cases harboring *BRAF*^V600E^ mutation versus PTC cases with wild-type BRAF (p = 0.042; Fig 6A).

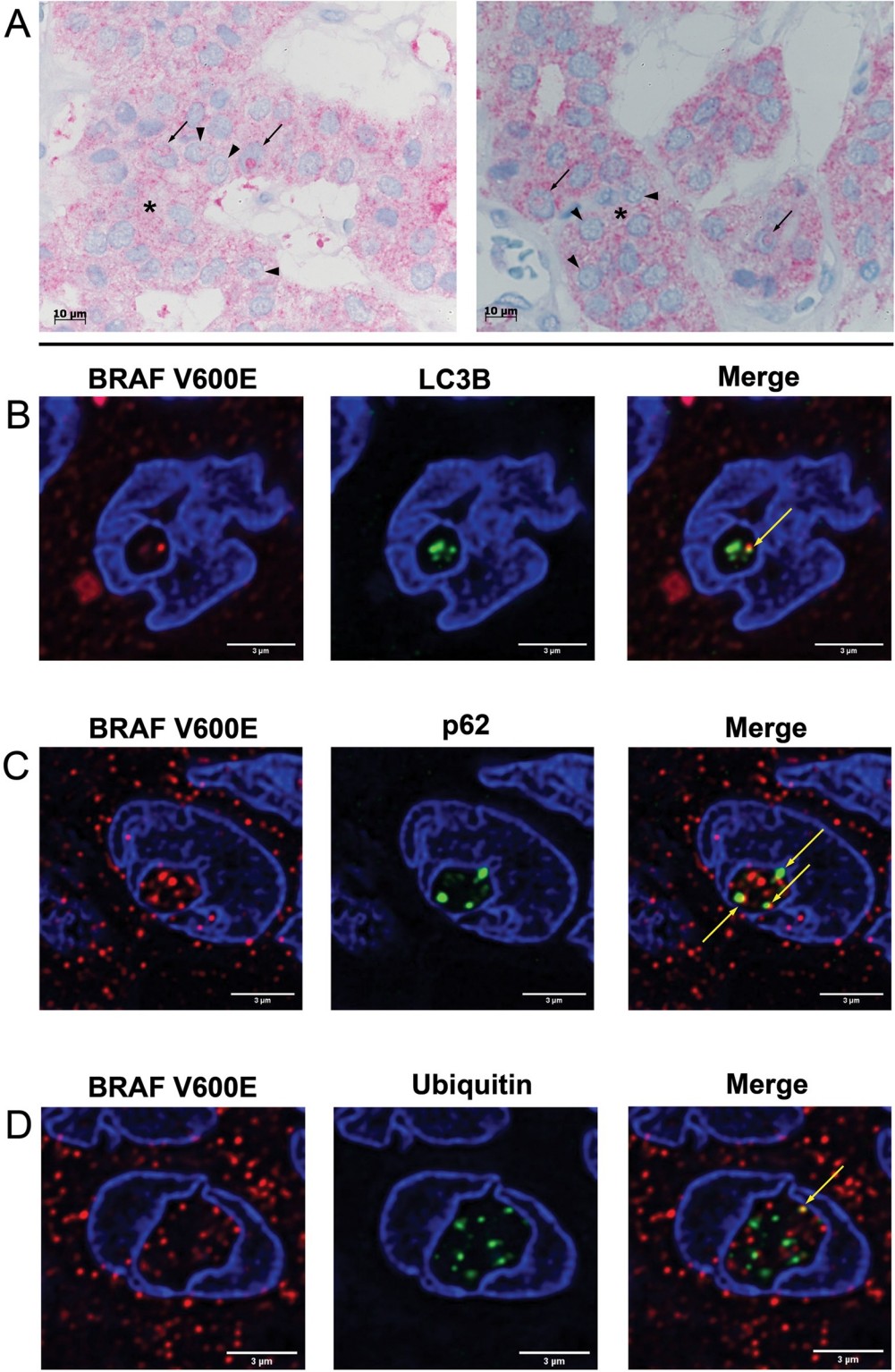

**Fig 5. Study of mutant BRAF protein found within the inclusions. (A)** BRAF^{V600E}-IHC analysis of PTC sections depicts additional to cytoplasmic immunopositivity (asterix) an accumulation of BRAF^{V600E} within the intranuclear inclusions (NI). The left image display NI with varying degrees of immunoreactivity for BRAF^{V600E} with very strong immunostaining within the NI (arrows) and weaker staining (arrowheads). The right image demonstrates NI containing an accumulation of BRAF^{V600E} (arrows) and NI with almost no immunoreactivity for mutant BRAF

(arrowheads). Original magnifications: 1,000 X. **(B-D)** Double-IF studies demonstrate co-localization of BRAF$^{V600E}$ with autophagy-associated proteins in NI of PTC sections. Nuclei were stained with DAPI (blue) to detect NI. **(B)** BRAF$^{V600E}$/LC3B double-IF labeling. The images reveal the same NI with immunopositivity for BRAF$^{V600E}$ (red staining) and for LC3B (green staining). The merged image shows the superimposition of the signals for BRAF$^{V600E}$ and LC3B with formation of yellow color (arrow) indicating BRAF$^{V600E}$/LC3B co-localization. **(C)** BRAF$^{V600E}$/p62 double-IF labeling. An NI with accumulation of BRAF$^{V600E}$ (red staining) and p62 (green staining) is shown; BRAF$^{V600E}$/p62 is demonstrated by intense yellow color (arrows) in the merged image. **(D)** BRAF$^{V600E}$/ubiquitin double-IF labeling. A nucleus with a large inclusion is shown with positive immunoreactivity for both mutant BRAF (red staining) and ubiquitin (green staining); the merged color yellow (arrow) points out the co-localisation of BRAF$^{V600E}$ and ubiquitin in the same inclusion.

## Correlation between p62, ubiquitin, LC3B, cathepsin B, cathepsin D expression and *BRAF* mutational status

One aim of this study was to investigate the reason of the occurrence of NI. As these inclusions mainly occurred in PTC cases, we focused on this tumor entity. We detected by Mann–Whitney U-test a significant correlation between immunoreactivity for autophagy-associated proteins in the inclusions and *BRAF* mutational status. Briefly, the number of NI with positive immunostaining for p62, ubiquitin, LC3B, cathepsin B and cathepsin D were significantly higher (p≤0.035) in PTC cases harboring *BRAF*$^{V600E}$ mutation versus PTC tumors with *BRAF* wild-type (Fig 6B–6F).

## Discussion

Intranuclear inclusions have been described both in normal and neoplastic tissues [1–7, 24]. The presence of NI is generally regarded as an unspecific morphologic feature. Nevertheless,

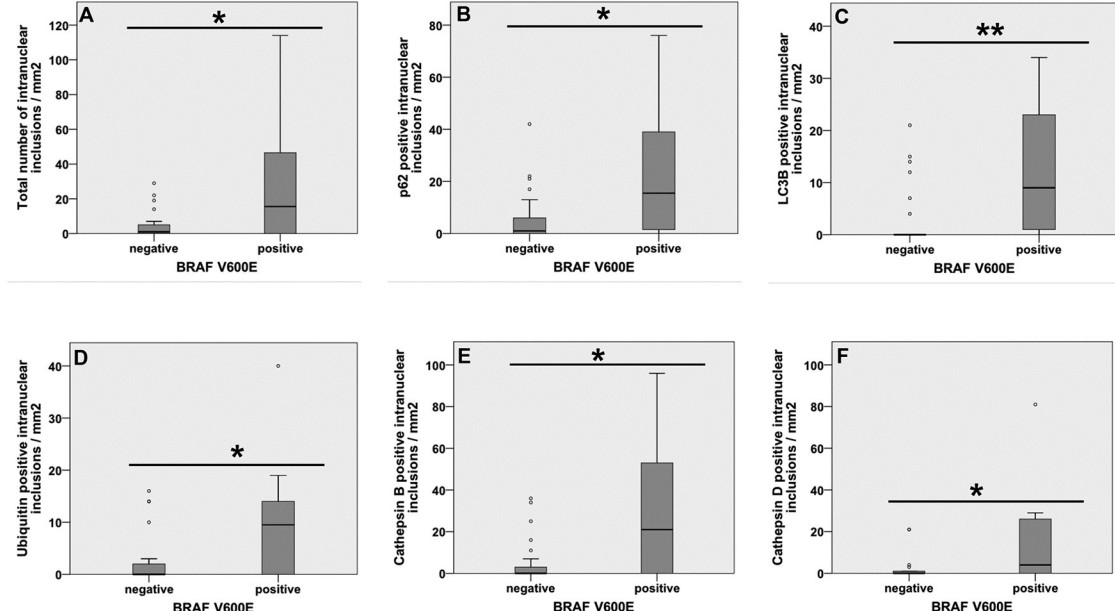

**Fig 6. Associations between *BRAF* mutational status and NI in PTCs. (A)** The diagram depicts a significant association between *BRAF* mutational status and the number of NI/case with p = 0.042. **(B-F)** Significant correlation between *BRAF* mutational status and immunoreactivity for autophagy-associated proteins in NI is shown. **(B)** p62 positive NI/case vs. *BRAF*$^{V600E}$: p = 0.024, **(C)** LC3B positive NI/case vs. *BRAF*$^{V600E}$: p = 0.003, **(D)** Ubiquitin positive NI/case vs. *BRAF*$^{V600E}$: p = 0.032, **(E)** Cathepsin B positive NI/case vs. *BRAF*$^{V600E}$: p = 0.035 and **(F)** Cathepsin D positive NI/case vs. *BRAF*$^{V600E}$: p = 0.013. Results are significant at *p≤0.05, **p≤0.01, and ***p≤0.001.

they are used as a diagnostic feature in several tumors such as melanomas and meningiomas and as a prognostic factor in clear cell renal cell carcinoma [10]. Also, they have a diagnostic significance in thyroid cancer [4, 5, 12]. Therefore, we investigated NI in thyroid cancer in more detail to clarify whether they also have a potential biological function.

NI are known to represent invaginations of the cytoplasm into the nucleus [10]. In thyroid carcinoma nuclear membrane-bounded inclusions of different size and shape are found [10]. Occasionally, artificially formed so-called "bubbles" without limiting nuclear membrane-resembling inclusions are observed [11]. This is in line with our results. Asioli et al used emerin staining for the detection of inclusions to exclude artificial bubbles [11]. In our present study, we have established diagnostic criteria for true NI. First, we have only considered an inclusion as positive if this inclusion was limited by a lamin AC (nuclear membrane marker) stained intact membrane proving that these inclusions are not artificial bubbles but real inclusions. Second, the NI should be completely closed in the plane seen through the microscope. Arora et al stated that the finding of inclusions of different shape and size in sections of the same tumor indicates a continuous process producing the inclusions [10]. Thus, we suppose there may be several stages of inclusion formation seen through the microscope. As we are interested in the stage when the invagination is completely closed we focused only on those inclusions, which were completely closed. Since the light microscope provides an image only in two dimensions we additionally performed 3D reconstructions of some nuclei.

Ultrastructural studies have shown that the inclusions in thyroid carcinoma cells are formed by the nuclear envelope [5, 12, 13]. We confirmed this, supported by our transmission electron microscope (TEM) images showing nuclei with NI and invaginations and both are limited by a nuclear membrane. We further investigated this topic and demonstrated by 3D reconstruction (Fig 3) that these NI were completely surrounded by the nuclear membrane and closed. Papotti et al also studied PTC nuclei but they reported that the inclusions are always connected to the cell cytoplasm [25]. This is in contrast to our results as our 3D reconstructions prove that the inclusions were completely located within the nucleus lacking any contact to the cytoplasm. It is very likely that these inclusions develop by invagination of the cytoplasm into the nucleus followed by later closure of the invagination. We suppose that the 3D reconstruction shown in Papotti et al reflects an early stage of a cytoplasmic invagination in the process of NI formation. However, nobody can exclude that some invaginations may not close and stay connected to the cytoplasm for a longer time.

TEM studies of Oyama et al showed that NI in thyroid carcinoma cells contain cell organelles, fragments of mitochondria and heterolysosomes [5], which is also in line with our TEM results (Fig 2). As they observed autophagosomes with crumpled membranes showing degradation in the inclusions, they discussed the possibility that these inclusions may be true inclusions and are not connected by the surrounding cytoplasm to explain the storage of products in the inclusions.

We investigated this observation in more detail and showed autophagy-associated proteins p62, ubiquitin and LC3B almost exclusively located in the NI with an accumulation of these proteins there but only weak cytoplasmic staining (Fig 1B) indicating it is not merely a passive invagination process. Our studies of inclusions using immunohistochemistry (Fig 1B), electron microscopy (Fig 2), and 3D reconstruction (Fig 3) showed that these inclusions are completely closed on the 2D and 3D plane. The content of these inclusions is more condensed and contains progressively degenerated cell organelles when compared to the organelles in the cytoplasm which supports again that a completely enclosed NI has no continuity into to the cytoplasm (Fig 2). NI with highly degenerative cytoplasmic components were already observed in the mid and late 1950s [6, 26]; they assumed that complete separation of the inclusions from the cytoplasm could induce a faster degeneration process within the inclusions. In an earlier

study [27] we detected that the lack of survivin in hepatocytes causes lack of chromosomal passenger complex proteins at the centromeres during mitosis, but reappeared in NI in interphase nuclei. The reason why these proteins are detected in NI is unclear, but it can be assumed that they could be degraded in the inclusions. Recently, we showed that NI in HCC contained autophagy-associated proteins [8]. In the current study we provide evidence that cathepsin B and cathepsin D (Fig 1B) are present inside NI. These proteases are usually restricted to lysosomes. In this context, we assume that they digest the content of the inclusions and thus promote the degeneration of the organelles [8]. The possibility of protein digestion in NI of meningiomas [9] and in nuclei with spherical eosinophilic inclusions [28] has already been discussed.

We were able to find p62 positive inclusions in 29.3%, ubiquitin in 20.4%, LC3B in 20%, cathepsin B in 20.6% and cathepsin D in 14.6% of all NI and a positive association of them to each other. We demonstrate by immunofluorescence double-staining that p62/ubiquitin, LC3B/p62 and ubiquitin/LC3B are co-localized in NI. This is consistent with other studies pointing out autophagy-associated proteins [8, 29] and autophagy activities [30, 31] in the nucleus. Intriguingly, our TEM results suggest different stages of degradation of included material in NI (Fig 2C). Nevertheless, additional functional experiments are necessary to confirm the hypothesis of autophagy in the NI.

We observed that the NI were mostly present in the PTCs and were nearly absent in other thyroid cancer entities, which is in line with the literature as they are a diagnostic criteria for PTC [21, 22]. For example, no NI were detected in the 3 ATC and 2 PDTC cases, harboring NRAS Q61R mutation. The presence of $BRAF^{V600E}$ mutation in 32% of PTCs confirms the literature, although our result is in the lower range, as it is reported that the $BRAF^{V600E}$ mutation occurs in about 45% (30 to 70%) of PTCs [32–36]. The occurrence of $BRAF^{V600E}$ mutation depends, among others, on the region, where the study is conducted and the available diet there [37]; Guan et al demonstrated the association of high iodine intake with $BRAF^{V600E}$ mutation [38]. Our study group consisted of patients from the European region, who do not consume iodine-rich seafood so often. In addition, the prevalence of $BRAF^{V600E}$ mutation is also depended on the studied variant of PTC. In our cohort were also 9 follicular variant PTCs and 4 solid variant PTCs, whereby PTC solid variant rarely harbors $BRAF^{V600E}$ mutation [39] and the $BRAF^{V600E}$ mutation rate is in PTC follicular variant with 12–40% lower than in PTC conventional and PTC tall cell variant [32, 40, 41]. In our current study, we found out that the number of NI was significantly higher in PTCs harboring a $BRAF^{V600E}$ mutation. This is in line with other studies on thyroid carcinomas [42–45]. They observed a positive correlation between $BRAF^{V600E}$ mutation and nuclear features of PTC (including NI) and tumor cells first defined by Finkelstein et al as "plumb eosinophilic cells" also showing inclusions. However, these studies focused on a general association between $BRAF^{V600E}$ mutation and the classic nuclear features of PTC and did not investigate explicitly membrane-limited NI, as we did. The NI we observed were morphologically similar to the "plump cells" described above, with the height of tumor cells less than twice the width and mostly eosinophilic [45].

Further, we asked whether $BRAF^{V600E}$ mutation plays a role in the formation of inclusions. It is documented that in human melanoma cells $BRAF^{V600E}$ induces spindle abnormalities, excess centrosomes and misaggregation of chromosomes leading to aneuploidy [46] due to phosphorylation of MPS1 by $BRAF^{V600E}$ [47]. Fischer et al [48] reviewed the reasons for alterations in the nuclear envelope (NE) in human cancers. They reported that inter alia chromosomal instability and aneuploidy can lead to alterations in the NE such as cell-to-cell variation in NE size and shape and to deep infoldings. Further, Fischer et al showed that $RET/PTC1$ microinjection in normal human thyroid epithelial cells induced NE irregularity during the

interphase and the forming of inclusions [49, 50]; they concluded that *BRAF* mutations may directly lead to inclusions by altering NE and chromatin organization [48]. This is supported by a study showing that targeted expression of BRAF^V600E in thyroid cells of transgenic mice resulted in irregularity of nuclear contours and occasional nuclear groves [51, 52]. The mechanisms involved in the formation of inclusions are, despite of several studies [53–55], still quite unknown. We suppose that different factors may induce the formation of inclusions and that the *BRAF^V600E* mutation is one of them.

Intriguingly, BRAF^V600E protein staining was also positive in NI. The localization of the BRAF^V600E antibody (VE1) in the nucleus, cytoplasm and cell membrane is documented (https://www.uniprot.org/uniprot/P15056). We scored the inclusions as positively stained only with simultaneous cytoplasmic staining and confirmation of the *BRAF^V600E* mutation by genetic analysis to exclude artificially incorrect results. We demonstrate the co-localization of BRAF^V600E/ubiquitin, BRAF^V600E/LC3B and BRAF^V600E/p62 in the NI by double-immunofluorescence. To our knowledge, this is the first study showing the presence of mutant BRAF simultaneous with autophagy-associated proteins in NI. Studies investigating the degradation of mutant BRAF revealed that BRAF^V600E requires the Hsp90 chaperone for stability and is degraded in response to Hsp90 inhibitors by ubiquitin-dependent degradation in the proteasome [56]. Further, as shown in thyroid cancer cells harboring *BRAF^V600E* mutation, the binding of the co-chaperone BAG3 to BRAF protected it from degradation by inhibiting HSP70-mediated delivery to the proteasome [57, 58]. Samant et al revealed that the E3 ubiquitin ligase CUL5 is involved in the Hsp90-inhibitor-induced BRAF^V600E degradation and NEDD8 conjugation of CUL5 is required for later degradation [59]. Recently, it was reported that the deubiquitinating enzyme USP28 stabilizes the F-box WD repeat-containing protein 7 (FBW7) and this results in SKP1/CUL1/F-box (SCF) mediated proteasomal degradation of BRAF^V600E [60, 61]. In the present study, we considered whether co-localization of mutant BRAF with autophagy-associated proteins is a passive process due to passive invagination of the cytoplasm into the inclusion, or rather an active proteolytic process. Although we also found few co-localizations of mutant BRAF with ubiquitin and LC3B (very weak staining) in the cytoplasm, a possible co-localization with p62 was lacking (S1 Fig). The immunoreactivity for ubiquitin and LC3B within the inclusions was much more pronounced than in the cytoplasm, with an accumulation of these proteins in NI (S1 Fig), which indicates an active proteolytic process. Thus, our results suggest that at least a part of the BRAF^V600E protein is degraded inside NI possibly through ubiquitin-dependent lysosomal degradation.

We found that *BRAF^V600E* positivity was first associated with an increase in number of inclusions and second with immunoreactivity for autophagy-associated proteins in the inclusions. The induction of autophagy in melanomas by hyperactivation of oncogenic BRAF is documented [18]. In papillary thyroid carcinoma *BRAF*-activated long non-coding RNA contributed to cell proliferation and activated autophagy [19]. Kim et al also reported the association of *BRAF^V600E* mutation with higher levels of autophagy-related proteins in PTC [62].

In summary, this study shows the existence of NI in thyroid carcinoma, which are completely surrounded by nuclear membrane with no contact to the cytoplasm. Therefore we would not term these inclusions "pseudoinclusions" but rather "real inclusions" or "true inclusions". How these true inclusions are formed is unknown but they may develop by closure of cytoplasmic invaginations. The presence of autophagy-associated proteins within the inclusions together with degenerated organelles and lysosomal proteases like cathepsins suggest that these inclusions play a role in autophagy and proteolysis. We demonstrate that the number of these true inclusions is positively associated with the *BRAF^V600E* mutation and show that mutant BRAF is detectable within these inclusions.

Double-immunofluorescence revealed mutant BRAF is co-localized with p62, LC3B and ubiquitin, respectively, which may imply that mutant BRAF is exposed to a proteolytic process.

Our results suggest that NI are not only of diagnostic importance in thyroid cancer, but above all may have a potential biological function.

## Supporting information

**S1 Fig. Double-IF studies of BRAF^V600E protein in PTC sections.** (A-B) BRAF^V600E/ubiquitin double IF labelling (A) The image demonstrats co-localization of mutant BRAF (red) with ubiquitin (green) within the inclusions (arrow) proved by the merged color yellow with lack of co-localization in the cytoplasm (B) Another nucleus with inclusions containing an accumulation of ubiquitin is seen with co-localization of BRAFV600E/ubiquitin both within the intranuclear inclusion (NI) and in the cytoplasm (arrows). Both the merged color yellow and the immunoreactivity for ubiquitin are stronger in the NI than in the cytoplasm; only weak distribution of ubiquitin is seen in the cytoplasm. (C-D) After z-stack analysis of the same nucleus BRAF^V600E/LC3B co-localizations were found on two different z-planes (C) Co-localization of BRAF^V600E (red) with LC3B (green) in the NI (arrow) but not in the cytoplasm is shown in this single z-plane image. (D) Few co-localizations of mutant BRAF with LC3B are seen in the cytoplasm (arrows) with no co-localization within the same NI on the other z-plane. Immunostaining for LC3B was stronger in the NI than in the cytoplasm demonstrating an accumulation of LC3B within the NI. (E-F) BRAF^V600E/p62 double IF labelling reveals co-localization of BRAF^V600E (red) with p62 (green) in two inclusions proven by the merged color yellow (arrows) with lack of co-localization in the cytoplasm.
(TIF)

**S1 Table. Immunohistochemistry antibodies and staining protocols.**
(PDF)

**S2 Table. NGS study: Panel of analyzed genes and exons.**
(PDF)

**S3 Table. 3D-Imaging of the inclusions: Antibodies used for double-immunofluorescence and staining conditions.**
(PDF)

**S4 Table. Double-labeling immunofluorescence microscopy: LC3B/ubiquitin, p62/ ubiquitin and LC3B/p62.**
(PDF)

**S5 Table. Double-labeling immunofluorescence microscopy: BRAF^V600E/LC3B, BRAF^V600E/ p62 and BRAF^V600E/ubiquitin.**
(PDF)

**S6 Table. NGS study: Mutations in the analyzed genes of the thyroid carcinoma cohort.**
(PDF)

## Acknowledgments

We thank Dorothe Möllmann, Martin Schlattjan and Laura Malkus for their excellent technical assistance. In addition, Dorothe Möllmann developed the method for isolation of nuclei with subsequent immunostaining, which is available at: http://dx.doi.org/10.17504/protocols. io.78phrvn.

## Author Contributions

**Conceptualization:** Suzan Schwertheim, Hideo A. Baba, Kurt W. Schmid.

**Data curation:** Thomas Herold, Christoph M. Schaefer.

**Formal analysis:** Suzan Schwertheim, Holger Jastrow, Hideo A. Baba.

**Investigation:** Suzan Schwertheim, Sarah Theurer, Holger Jastrow, Thomas Herold, Saskia Ting, Daniela Westerwick, Stefanie Bertram, Christoph M. Schaefer, Julia Kälsch, Hideo A. Baba, Kurt W. Schmid.

**Methodology:** Suzan Schwertheim, Sarah Theurer, Holger Jastrow, Thomas Herold, Saskia Ting, Daniela Westerwick, Stefanie Bertram, Christoph M. Schaefer, Julia Kälsch, Hideo A. Baba.

**Project administration:** Suzan Schwertheim, Hideo A. Baba, Kurt W. Schmid.

**Supervision:** Suzan Schwertheim, Hideo A. Baba, Kurt W. Schmid.

**Writing – original draft:** Suzan Schwertheim, Hideo A. Baba.

**Writing – review & editing:** Suzan Schwertheim, Sarah Theurer, Holger Jastrow, Thomas Herold, Saskia Ting, Daniela Westerwick, Stefanie Bertram, Christoph M. Schaefer, Julia Kälsch, Hideo A. Baba, Kurt W. Schmid.

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
