## [Decision Letter · Decision Letter 0]

18 Sep 2019

PONE-D-19-21366

New insights into intranuclear inclusions in thyroid carcinoma: association with autophagy and with BRAFV600E mutation

PLOS ONE

Dear Dr. Schwertheim,

Thank you for submitting your manuscript to PLOS ONE. After careful consideration, we feel that it has merit but does not fully meet PLOS ONE’s publication criteria as it currently stands. Therefore, we invite you to submit a revised version of the manuscript that addresses the points raised during the review process.

We would appreciate receiving your revised manuscript by Nov 02 2019 11:59PM. To enhance the reproducibility of your results, we recommend that if applicable you deposit your laboratory protocols in protocols.io, where a protocol can be assigned its own identifier (DOI) such that it can be cited independently in the future. For instructions see: http://journals.plos.org/plosone/s/submission-guidelines#loc-laboratory-protocols

We look forward to receiving your revised manuscript.

Kind regards,

Francis Moore, Jr.

Academic Editor

PLOS ONE

Journal Requirements:

2. We noticed minor instances of text overlap with the following previous publication(s), which need to be addressed:

doi: 10.1002/cjp2.129

In your revision please ensure you cite all your sources (including your own works), and quote or rephrase any duplicated text outside the methods section. Further consideration is dependent on these concerns being addressed.

Additional Editor Comments:

Please address the comments of the two reviewers.

(1) I think the authors samples maybe slightly skewed however and they may need to address the prevalence of BRAF in the literature that occurs in classical, poorly differentiated and FVTC. Rates in most studies of BRAF positive at in the 60-70% range. Can the authors explain their sample selection and why they were unable to see BRAF mutations in these other subgroups. There is a need to increase your sample size to explain the rational for not observing these changes as it would mean the clinical sample size is too low.

I note the authors correlate the autophagy genes in patients with the BRAF mutation. There are 14 patients with the BRAF mutation and a significantly higher number of tumor samples with at least upregulation of one or more of the autophagy genes. Can the authors comment on the total cumulative number of patients with at least one of the autophagy genes unregulated. Can the authors also explain if the BRAF mutation is only present in 14% of patients and more than 29% of patients have a change in at least one autophagy gene the effect is only at best 50% of patients with changes in the autophagy genes. And what is different about these tumors and the tumor biology versus the patients without these features. Potentially a correlation with recurrence and disease free survival and outcomes would be important.

It would be also important to see the labeling of BRAF wt in the inclusions. I think the authors need to see if there is clustering of wt in nuclear inclusions.

(2) There are minor editorial corrections/suggestions:

1. Line 79 take out word "briefly".

2. Line 204 Did you really homogenize the cells in 70% ethanol (more likely aquaeous)? Also, cells should not have needed permeabilization since they went through xylene.

3. Line 318-19. The writing is not clear. I think you mean "No other genes were mutated in more than one case".

4. Line 324. Not clear. I think you can delete the beginning of the sentence to say "Lamin A/C positive inclusions occur most frequently..."

5. Line 389-390. I don't think you have any basis to speculate about the temporal aspects of the fate of inclusions. I would say something like "as if it could become ejected soon"

6. The authors may want to include this reference on intranuclear cytoplasmic inclusions:

Rezk S, Brynes RK, Nelson V, Thein M, Patwardhan N, Fischer A, Khan A.

2004. beta-Catenin expression in thyroid follicular lesions: Potential role in

nuclear envelope changes in papillary carcinomas. Endocr Pathol 15:329–

337.

Reviewers' comments:

Reviewer's Responses to Questions

**Comments to the Author**

1. Is the manuscript technically sound, and do the data support the conclusions?

Reviewer #1: Yes

Reviewer #2: Yes

2. Has the statistical analysis been performed appropriately and rigorously? 

Reviewer #1: Yes

Reviewer #2: Yes

3. Have the authors made all data underlying the findings in their manuscript fully available?

Reviewer #1: Yes

Reviewer #2: Yes

4. Is the manuscript presented in an intelligible fashion and written in standard English?

Reviewer #1: Yes

Reviewer #2: No

5. Review Comments to the Author

Reviewer #1: I commend the authors of the study for their findings. Their correlation of autophagy genes

I think the authors samples maybe slightly skewed however and they may need to address the prevalence of BRAF in the literature that occurs in classical, poorly differentiated and FVTC. Rates in most studies of BRAF positive at in the 60-70% range. Can the authors explain their sample selection and why they were unable to see BRAF mutations in these other subgroups. There is a need to increase your sample size to explain the rational for not observing these changes as it would mean the clinical sample size is too low.

I note the authors correlate the autophagy genes in patients with the BRAF mutation. There are 14 patients with the BRAF mutation and a significantly higher number of tumor samples with at least upregulation of one or more of the autophagy genes. Can the authors comment on the total cumulative number of patients with at least one of the autophagy genes unregulated. Can the authors also explain if the BRAF mutation is only present in 14% of patients and more than 29% of patients have a change in at least one autophagy gene the effect is only at best 50% of patients with changes in the autophagy genes. And what is different about these tumors and the tumor biology versus the patients without these features. Potentially a correlation with recurrence and disease free survival and outcomes would be important.

It would be also important to see the labeling of BRAF wt in the inclusions. I think the authors need to see if there is clustering of wt in nuclear inclusions.

Reviewer #2: This manuscript has surprising, convincing, important new data on the structural features of intranuclear cytoplasmic inclusions: Intranuclear cytoplasmic inclusions are (at least in the carefully studied examples) completely bound by nuclear lamina and nuclear membranes.

There are minor editorial corrections/suggestions:

1. Line 79 take out word "briefly".

2. Line 204 Did you really homogenize the cells in 70% ethanol (more likely aquaeous)? Also, cells should not have needed permeabilization since they went through xylene.

3. Line 318-19. The writing is not clear. I think you mean "No other genes were mutated in more than one case".

4. Line 324. Not clear. I think you can delete the beginning of the sentence to say "Lamin A/C positive inclusions occur most frequently..."

5. Line 389-390. I don't think you have any basis to speculate about the temporal aspects of the fate of inclusions. I would say something like "as if it could become ejected soon"

6. The authors may want to include this reference on intranuclear cytoplasmic inclusions:

Rezk S, Brynes RK, Nelson V, Thein M, Patwardhan N, Fischer A, Khan A.

2004. beta-Catenin expression in thyroid follicular lesions: Potential role in

nuclear envelope changes in papillary carcinomas. Endocr Pathol 15:329–

337.

6. PLOS authors have the option to publish the peer review history of their article (what does this mean?). If published, this will include your full peer review and any attached files.

Reviewer #1: No

Reviewer #2: Yes: Andrew H. Fischer

---

## [Author Response · Author response to Decision Letter 0]

25 Oct 2019

Editor Comments: First of all, we want to thank you for taking the time to review our manuscript. We have highlighted the changes in our manuscript within the document by using coloured (blue) text; deleted text is marked in red and crossed out. The information regarding page numbers refers to the previous version of the manuscript. Journal Requirements (JR) and the responds of the author (AC) are listed below. 

Journal Requirements:

JR.1. When submitting your revision, we need you to address these additional requirements.

Please ensure that your manuscript meets PLOS ONE's style requirements, including those for file naming. The PLOS ONE style templates can be found at http://www.journals.plos.org/plosone/s/file?id=wjVg/PLOSOne_formatting_sample_main_body.pdf and http://www.journals.plos.org/plosone/s/file?id=ba62/PLOSOne_formatting_sample_title_authors_affiliations.pdf

AC.1. We have checked PLOS ONE's style requirements and we have incorporated corresponding author’s initials in parentheses after the email address. Additionally, we have corrected the formatting of the headings.

JR.2. We noticed minor instances of text overlap with the following previous publication(s), which need to be addressed:

doi: 10.1002/cjp2.129

In your revision please ensure you cite all your sources (including your own works), and quote or rephrase any duplicated text outside the methods section. Further consideration is dependent on these concerns being addressed.

AC.2. Thank you for your comment regarding minor instances of text overlap with our own previous publication. We have checked this and found some parts mainly concerning background information and some idioms and we apologize for this. We rephrased or quoted any duplicated text outside the methods section in the revised manuscript.

Response to the Reviewer: First of all, we want to thank you for taking the time to review our manuscript. We have highlighted the changes in our manuscript within the document by using colored (blue) text; deleted text is marked in red and crossed out. The information regarding page numbers refers to the previous version of the manuscript. The comments of the reviewer (RC) and the responds of the author (AC) are listed below. 

Response to Reviewer 1 

RC: I think the authors samples maybe slightly skewed however and they may need to address the prevalence of BRAF in the literature that occurs in classical, poorly differentiated and FVTC. Rates in most studies of BRAF positive at in the 60-70% range. Can the authors explain their sample selection and why they were unable to see BRAF mutations in these other subgroups. There is a need to increase your sample size to explain the rational for not observing these changes as it would mean the clinical sample size is too low.

AC: Thank you for pointing out the prevalence of BRAF in the literature regarding the different entities. We incorporated additional references into the discussion section (line 612 of the previous manuscript) of the revised manuscript. It is documented that the occurrence of BRAF V600E mutation depends, among others, on the region, where the study is conducted and the available diet there [1]. Guan et al demonstrated the association of high iodine intake with BRAF V600E mutation [2]. The study cohort consisted of patients with classical PTC and revealed that in regions with high iodine contents in natural drinking water the prevalence of BRAF V600E mutation was significantly higher (69%) than in regions with normal iodine content (53%) [2]. A study of Jin et al revealed that 63.7% of the Chinese patients with PTC harbored the BRAF V600E mutation and they pointed out that high dietary iodine intake from rich sea foods may contribute to this high prevalence of BRAF V600E mutation; however they did not elaborate on the variants of studied PTCs. The prevalence of BRAF V600E mutation is also depended on the studied variant of PTC. It is documented that BRAF V600E mutation occurs significantly more frequently in the conventional variant of PTC than in the follicular variant with 58% versus 31% [3] and 75% versus 40% [4]. Only classical PTC cases were analyzed in the above mentioned study by Guan et al. with the detection of 53% BRAF V600E mutation in regions with normal iodine content. There are also studies reporting a lower prevalence of BRAF V600E mutation in about 45% (30 to 70%) of PTCs [5–9], with 12% of follicular variant PTC and 77% of tall cell variant PTC harboring this mutation [5]. In our study cohort 14 of 43 PTCs (32%) harbored the BRAF V600E mutation. The reason for this relatively low occurrence of the BRAF V600E mutation could be that first of all, our study group consisted of patients from the European region, who do not consume iodine-rich seafood so often. Secondly, in our study group were also 9 follicular variant PTCs, 1 hobnail variant PTC, 1 columnar cell variant PTC and 4 solid variant PTCs, whereby PTC solid variant rarely harbors BRAF V600E mutation [10] but more frequently BRAF VK600-1E mutation [10, 11] and BRAF V600delinsAL [5, 10]. Our study cohort consisted of 10 ATCs, 10 PDTCs and 9 FTCs, which were all BRAF V600E negative. It is documented that PDTCs and ATCs were more likely to have BRAF V600E mutation when they arise from papillary carcinoma [9, 12, 13]. The BRAF V600E mutation was found in about 5-20% [14–16] of PDTCs and in 11-45% of ATCs [17–19]. In the study of Landa et al, in which 45% of ATCs harbored the BRAF V600E mutation, they used NGS technique and adopted an ultra-deep sequencing strategy, enabling an average depth of coverage [18, 20]. In our current study, we used also NGS but we did not perform ultra-deep sequencing described by Landa et al. Another reason why we detected no BRAF mutation in our PDTC and ATC cases might be that they developed de novo rather than arised from papillary carcinomas [9, 12]. Nevertheless, our current study did not focus on the prevalence of BRAF V600E mutation in thyroid cancer. Our aim was to show that NI so often observed in daily microscopic observations, are completely surrounded by the nuclear membrane. In addition, we wanted to show that these inclusions, which have so far been considered of less important, might have a function because they contain accumulations of proteins associated with autophagy and proteolysis. Therefore, although we appreciate your comment, we believe that increasing the number of ATC and PDTC cases would not be useful, as the message regarding the characteristics of intranuclear inclusions remains the same.

RC: I note the authors correlate the autophagy genes in patients with the BRAF mutation. There are 14 patients with the BRAF mutation and a significantly higher number of tumor samples with at least upregulation of one or more of the autophagy genes. Can the authors comment on the total cumulative number of patients with at least one of the autophagy genes unregulated. Can the authors also explain if the BRAF mutation is only present in 14% of patients and more than 29% of patients have a change in at least one autophagy gene the effect is only at best 50% of patients with changes in the autophagy genes. 

AC: We apologize that the manuscript was not clear at this point, which led to misunderstandings, as we did not investigate the expression of genes. We only found out that in 14-29% of all cases there was an accumulation of autophagy-associated proteins within the inclusions. Further, the number of NI, which showed immunopositivity for autophagy-associated proteins within the inclusions was higher in BRAF V600E positive cases than in cases with BRAF wild-type. We agree that about 50% of the cases harbored NI with an accumulation of autophagy-associated proteins within them although they have no BRAF mutation. But the number of such filled NI was lower in BRAF wild-type cases. In our opinion the BRAF V600E mutation is only one factor, which may regulate the accumulation of the autophagy-associated proteins within NI and the formation of NI (line 635-637 of the previous manuscript). Nevertheless, we assume that there are also other factors that play a role in these biological processes. 

RC: And what is different about these tumors and the tumor biology versus the patients without these features. Potentially a correlation with recurrence and disease free survival and outcomes would be important.

AC: Unfortunately, we don’t have survival data of the patients. 

RC: It would be also important to see the labeling of BRAF wt in the inclusions. I think the authors need to see if there is clustering of wt in nuclear inclusions.

AC: Thank you for your suggestion. We provided the antibody anti-BRAF (#NBP1-47668, Novus, Littleton, CO), which detects BRAF wild-type (https://www.novusbio.com/PDFs/NBP1-47668.pdf). 

We performed IHC studies on whole FFPE tissue sections of two PTCs of our study group: one case harboring the BRAF V600E mutation and the other case BRAF wild-type, proven formerly by NGS. We detected in both cases NI with accumulations of BRAF wild-type protein and NI without immunoreactivity for BRAF wild-type regardless of BRAF mutation status (Additional figure for review is provided below). Further, we observed varying degrees of immunoreactivity for BRAF wild-type within the NI (arrows) whereby some NI showed relatively weaker immunostaining (arrows). We found BRAF wild-type immunopositivity also in the case in which formerly BRAF V600E mutation was proved by NGS. This result is unexpected and can have several reasons. One explanation may be heterogeneity in this case with both BRAF wild-type and BRAF V600E cells coexisting in the same tumor. Also other studies have observed the existence of BRAF heterogeneity in some tumors [21, 22]. Another explanation can be that the mutated BRAF protein also reacts to some degree with this antibody. We contacted Novus Biologicals, from which the antibody anti-BRAF (#NBP1-47668) was purchased, and asked how specific this antibody is. The answer was that since this antibody is made from a full length recombinant, which is based on the wild type sequence, NP_004324 its possible that it may or may not pick up the mutated form. Additionally, they said that as long as the epitope of NBP1-47668 is not directly over the mutation in BRAF V600E then this antibody should pick up the wild-type and mutant form. 

In our manuscript we documented the accumulation of mutant-BRAF protein within NI and now we also found BRAF wild-type protein within NI. It is documented that high BRAF expression was significantly correlated with poor patient survival in melanoma [23, 24]. As we also found clustering of BRAF wild-type in NI this might due to a biological mechanism to deal with excess BRAF protein. 

Additional Figure for Review

Immunohistochemistry: BRAF Antibody (OTI5A9) [NBP1-47668] 

The image is displayed in the attached Word document.

The images depict immunostaining of representative PTC cases harboring BRAF wild-type (A, B) and BRAF V600E mutation (C, D) proven by NGS; varying degrees of immunoreactivity for BRAF (#NBP1-47668, Novus, Littleton, CO) is shown both in the intranuclear inclusions (arrows) and in the cytoplasm. Original magnifications: 1,000 X. 

Response to the Reviewer: First of all, we want to thank you for taking the time to review our manuscript. We have highlighted the changes in our manuscript within the document by using colored (blue) text; deleted text is marked in red and crossed out. The information regarding page numbers refers to the previous version of the manuscript. The comments of the reviewer (RC) and the responds of the author (AC) are listed below. 

Response to Reviewer 2

RC: 1. Line 79 take out word "briefly".

AC: 1. Thank you for this point; we have deleted it. 

RC: 2a. Line 204 Did you really homogenize the cells in 70% ethanol (more likely aquaeous)? 

AC: 2a. Thank you very much for your comment. We agree with your assessment and apologize for this mistake. We missed to incorporate that after the rehydration-step with 70% ethanol, we homogenized the cells in Target Retrievel Solution pH9 (#S2367, Dako, Glostrup, Denmark). We rewrote the section “3D imaging of immunofluorescence-labeled isolated nuclei” and added it as a separate laboratory protocol with its own identifier (DOI) to our manuscript (http://dx.doi.org/10.17504/protocols.io.78phrvn). 

RC: 2b. Also, cells should not have needed permeabilization since they went through xylene.

AC. 2b. This is a very interesting point. We agree with you that permeabilization is not always needed as xylene interacts with the cell membrane and changes the permeability [25, 26]. Nevertheless, it is documented that the exact mechanism is not fully understood [25]. We are of the opinion that it also depends on the thickness of the tissue sections. To isolate complete nuclei, paraffin-embedded tissue sections were cut relatively thick (60 μm) to keep the nucleus intact as effectively as possible. In this case addition of xylene was performed to remove paraffin from the paraffin-embedded tissues. Since the mechanism of how xylene interacts with the cell membrane is not fully understood, we think it is better to add saponin to the Dako REAL Antibody Diluent (#S2022, Dako) to ensure that the antibody penetrates the nucleus. However, regarding our double-labeling immunofluorescence studies of FFPE tissue sections cut at 1 μm, experiments were carried out according to standard protocols and we did not add saponin to the antibody dilution buffer (Dako REAL Antibody Diluent, #S2022, Dako). In the troubleshooting data of abcam it is recommended if the antibody cannot penetrate the nucleus, where the protein is located (nuclear protein) to add a strong permeabilizing agent like Triton™ X-100 to the blocking buffer and antibody dilution buffer (https://www.abcam.com/content/immunohistochemistry-the-complete-guide). We permeabilized the cells with saponin according to a modified flow cytometry and cell sorting protocol [27]. We incubated the antibodies in the presence of saponin because it is documented that in contrast to many other detergents cell permeabilization with saponin is a reversible process and once saponin is removed from the sample, the pores of the membrane will be closed [28]. Saponin intercalates in the membrane structure, replacing cholesterol and leaves much of the membrane structure intact [29–31].

RC: 3. Line 318-19. The writing is not clear. I think you mean "No other genes were mutated in more than one case".

AC: 3. We agree that this part is unclear. Unfortunately, we cannot replace it by your words because we found for example in ARID1B gene the R1534K and R1552K mutations in 3 /107 (2.7 %) cases (supporting information S6 Table). These are mutations, which we did not want to discuss in our manuscript in more detail. For more openness and transparency, however, we have replaced the former text (line 318-19 of the previous manuscript) as follows: “We detected with a maximum of 3/107 (2.8%) cases a rather low incidence of mutations in the genes of the Wnt pathway and we did not find any associations between these mutations and the occurrence of NI.” 

RC: 4. Line 324. Not clear. I think you can delete the beginning of the sentence to say "Lamin A/C positive inclusions occur most frequently..."

AC: 4. We have deleted the beginning of the sentence.

RC: 5. Line 389-390. I don't think you have any basis to speculate about the temporal aspects of the fate of inclusions. I would say something like "as if it could become ejected soon"

AC: 5. We agree with you and we have replaced the section “which seems to be pushed to the edge of the nucleus for being ejected soon.” by “as if it could become ejected soon."

RC: 6. The authors may want to include this reference on intranuclear cytoplasmic inclusions:

Rezk S, Brynes RK, Nelson V, Thein M, Patwardhan N, Fischer A, Khan A. 2004. beta-Catenin expression in thyroid follicular lesions: Potential role in nuclear envelope changes in papillary carcinomas. Endocr Pathol 15:329– 337.

AC: 6. We have added to the introduction section (line 99 of the previous manuscript): “In addition, it is also documented that strong immunopositivity for ß-catenin was detected within NI in 83% of PTCs [32]. ”

References

1. Jin L, Chen E, Dong S, Cai Y, Zhang X, Zhou Y et al. BRAF and TERT promoter mutations in the aggressiveness of papillary thyroid carcinoma: a study of 653 patients. Oncotarget. 2016;7:18346-18355.

2. Guan H, Ji M, Bao R, Yu H, Wang Y, Hou P et al. Association of high iodine intake with the T1799A BRAF mutation in papillary thyroid cancer. J Clin Endocrinol Metab. 2009;94:1612-1617.

3. Smith RA, Salajegheh A, Weinstein S, Nassiri M, Lam AK-y. Correlation between BRAF mutation and the clinicopathological parameters in papillary thyroid carcinoma with particular reference to follicular variant. Hum Pathol. 2011;42:500-506.

4. Lim JY, Hong SW, Lee YS, Kim B-W, Park CS, Chang H-S et al. Clinicopathologic implications of the BRAF V600E mutation in papillary thyroid cancer: a subgroup analysis of 3130 cases in a single center. Thyroid. 2013;23:1423-1430.

5. Chiosea S, Nikiforova M, Zuo H, Ogilvie J, Gandhi M, Seethala RR et al. A novel complex BRAF mutation detected in a solid variant of papillary thyroid carcinoma. Endocr Pathol. 2009;20:122-126.

6. Cohen Y, Xing M, Mambo E, Guo Z, Wu G, Trink B et al. BRAF mutation in papillary thyroid carcinoma. J Natl Cancer Inst. 2003;95:625-627.

7. Katoh H, Yamashita K, Enomoto T, Watanabe M. Classification and general considerations of thyroid cancer. Ann Clin Pathol. 2015;3:1045.

8. Kimura ET, Nikiforova MN, Zhu Z, Knauf JA, Nikiforov YE, Fagin JA. High prevalence of BRAF mutations in thyroid cancer: genetic evidence for constitutive activation of the RET/PTC-RAS-BRAF signaling pathway in papillary thyroid carcinoma. Cancer Res. 2003;63:1454-1457.

9. Nikiforova MN, Kimura ET, Gandhi M, Biddinger PW, Knauf JA, Basolo F et al. BRAF mutations in thyroid tumors are restricted to papillary carcinomas and anaplastic or poorly differentiated carcinomas arising from papillary carcinomas. J Clin Endocrinol Metab. 2003;88:5399-5404.

10. Giorgadze TA, Scognamiglio T, Yang GCH. Fine-needle aspiration cytology of the solid variant of papillary thyroid carcinoma: A study of 13 cases with clinical, histologic, and ultrasound correlations. Cancer Cytopathol. 2015;123:71-81.

11. Trovisco V, Soares P, Soares R, Magalhaes J, Sa-Couto P, Sobrinho-Simoes M. A new BRAF gene mutation detected in a case of a solid variant of papillary thyroid carcinoma. Hum Pathol. 2005;36:694-697.

12. Schmid KW. Molecular pathology of thyroid tumors. Pathologe. 2010;31:229-233.

13. Soares P, Lima J, Preto A, Castro P, Vinagre J, Celestino R et al. Genetic alterations in poorly differentiated and undifferentiated thyroid carcinomas. Current genomics. 2011;12:609-617.

14. Dettmer M, Schmitt A, Komminoth P, Perren A. Poorly differentiated thyroid carcinoma. Pathologe. 2019;https://doi.org/10.1007/s00292-019-0600-9

15. Eloy C, Ferreira L, Salgado C, Soares P, Sobrinho-Simoes M. Poorly differentiated and undifferentiated thyroid carcinomas. Turkish Journal of Pathology. 2015;31

16. Nikiforov YE, Nikiforova MN. Molecular genetics and diagnosis of thyroid cancer. Nature Reviews Endocrinology. 2011;7:569-580.

17. Kunstman JW, Juhlin CC, Goh G, Brown TC, Stenman A, Healy JM et al. Characterization of the mutational landscape of anaplastic thyroid cancer via whole-exome sequencing. Hum Mol Genet. 2015;24:2318-2329.

18. Landa I, Ibrahimpasic T, Boucai L, Sinha R, Knauf JA, Shah RH et al. Genomic and transcriptomic hallmarks of poorly differentiated and anaplastic thyroid cancers. The Journal of clinical investigation. 2016;126:1052-1066.

19. Tiedje V, Ting S, Herold T, Synoracki S, Latteyer S, Moeller LC et al. NGS based identification of mutational hotspots for targeted therapy in anaplastic thyroid carcinoma. Oncotarget. 2017;8:42613-42620.

20. Xu B, Ghossein R. Genomic landscape of poorly differentiated and anaplastic thyroid carcinoma. Endocr Pathol. 2016;27:205-212.

21. Kurtulmus N, Ertas B, Saglican Y, Kaya H, Ince U, Duren M. BRAFV600E mutation: has it a role in cervical lymph node metastasis of papillary thyroid cancer. European thyroid journal. 2016;5:195-200.

22. Takata M. Identifying BRAF and KIT mutations in melanoma. Expert Review of Dermatology. 2013;8:171-176.

23. Bhandaru M, Safaee Ardekani G, Zhang G, Martinka M, McElwee KJ, Li G et al. A combination of p300 and Braf expression in the diagnosis and prognosis of melanoma. BMC cancer. 2014;14:398.

24. Safaee Ardekani G, Jafarnejad SM, Khosravi S, Martinka M, Ho V, Li G. Disease progression and patient survival are significantly influenced by BRAF protein expression in primary melanoma. Br J Dermatol. 2013;169:320-328.

25. Niaz K, Bahadar H, Maqbool F, Abdollahi M. A review of environmental and occupational exposure to xylene and its health concerns. EXCLI journal. 2015;14:1167-1186.

26. Tahti H. Neurotoxicity of organic solvents, studied with in vitro models. Alternatives to laboratory animals: ATLA. 1992

27. Radbruch A. Flow cytometry and cell sorting. Springer; 2000

28. Albu DI, Califano D, Avram D. Flow cytometry analysis of transcription factors in T lymphocytes Transcription Factors. Springer; 2010:377–390.

29. Castronovo V, Taraboletti G, Sobel ME. Functional domains of the 67-kDa laminin receptor precursor. J Biol Chem. 1991;266:20440-20446.

30. Willingham MC, Pastan IH. Atlas of immunofluorescence in cultured cells. Academic Press; 1985

31. Willingham MC. Fluorescence labeling of intracellular antigens of attached or suspended tissue-culture cells Immunocytochemical Methods and Protocols. Springer; 2010:153–164.

32. Rezk S, Brynes RK, Nelson V, Thein M, Patwardhan N, Fischer A et al. Beta-Catenin expression in thyroid follicular lesions: Potential role in nuclear envelope changes in papillary carcinomas. Endocr Pathol. 2004;15:329-337.

---

## [Editor Report · Decision Letter 1]

22 Nov 2019

New insights into intranuclear inclusions in thyroid carcinoma: association with autophagy and with BRAF V600E mutation

PONE-D-19-21366R1

Dear Dr. Schwertheim,

We are pleased to inform you that your manuscript has been judged scientifically suitable for publication and will be formally accepted for publication once it complies with all outstanding technical requirements.

With kind regards,

Francis Moore, Jr.

Academic Editor

PLOS ONE
---

## [Editor Report · Acceptance letter]

2 Dec 2019

PONE-D-19-21366R1 

New insights into intranuclear inclusions in thyroid carcinoma: association with autophagy and with BRAF V600E mutation 

Dear Dr. Schwertheim:

I am pleased to inform you that your manuscript has been deemed suitable for publication in PLOS ONE. Congratulations! Your manuscript is now with our production department. 

With kind regards,

on behalf of

Dr. Francis Moore, Jr. 

Academic Editor

PLOS ONE